# Compatibility of a Competition Model for Explaining Eye Fixation Durations During Free Viewing

**DOI:** 10.3390/e27101079

**Published:** 2025-10-18

**Authors:** Carlos M. Gómez, María A. Altahona-Medina, Gabriela Barrera, Elena I. Rodriguez-Martínez

**Affiliations:** Human Psychobiology Laboratory, Experimental Psychology Department, School of Psychology, University of Seville, 41018 Sevilla, Spain; maraltmed@alum.us.es (M.A.A.-M.); marbarrui@alum.us.es (G.B.); elisroma@us.es (E.I.R.-M.)

**Keywords:** eye fixation durations, saccades, competition model, exGaussian model, refractory period

## Abstract

Inter-saccadic times or eye fixation durations (EFDs) are relatively stable at around 250 ms, equivalent to four saccades per second. However, the mean and standard deviation are not sufficient to describe the frequency histogram distribution of EFD. The exGaussian has been proposed for fitting the EFD histograms. The present report tries to adjust a competition model (C model) between the saccadic and the fixation network to the EFD histograms. This model is at a rather conceptual level (computational level in Marr’s classification). Both models were adjusted to EFD from an open database with data of 179,473 eye fixations. The C model showed to be able, along with exGaussian model, to be compatible with explaining the EFD distributions. The two parameters of the C model can be ascribed to (i) a refractory period for new saccades modeled by a sigmoid equation (A parameter), while (ii) the ps parameter would be related to the continuous competition between the saccadic network related to the saliency map and the eye fixation network, and would be modeled through a geometric probability density function. The model suggests that competition between neural networks would be an organizational property of brain neural networks to facilitate the decision process for action and perception. In the visual scene scanning, the C model dynamic justifies the early post-saccadic stability of the foveated image, and the subsequent exploration of a broad space in the observed image. The code to extract the data and to run the model is added in the Appendix A. Additionally, entropy of EFD is reported.

## 1. Introduction

The primary function of eye movements is to position objects of interest within the visual field. The reception of visual information is most effective in the fovea, a highly sensitive region of the retina. The ocular motor system is specifically designed to keep objects of interest in focus in this area. Saccadic eye movements permit tracking in a fast manner and focus in the fovea certain parts of an image for intense visual scrutiny during eye fixations.

Saccadic movements are fast and precise, reaching speeds of up to 900°/s in humans. Their purpose is to redirect the gaze efficiently to a new visual stimulus. There is a linear relationship between the amplitude and duration of the saccades in the range between 1° and 50° [1,2,3], as well as between the maximum velocity and the amplitude of the movement, up to saccades with amplitudes of about 15°, at which eye velocity slowly saturates [4]. Due to their high speed, these movements are considered ballistic, i.e., their control depends on a previous calculation of their parameters, without feedback during execution. However, a moderate modulation of trajectory can be obtained in flight by visual inputs [5]. Once a saccade is completed, the eyes remain in a fixed position until a new movement is made. During this fixation period, the eyes exhibit small involuntary movements of three different types: tremor, drifting and microsaccades [6].

### 1.1. Short Description of Neural Control of Saccadic Eye Movement and Eye Fixations

The neural control of saccadic movement is complex and implicates a great number of interconnected structures. During saccadic movements directed toward the periphery (abduction), motoneurons and interneurons of the abducens nucleus increase their activity. Conversely, during inward movements (adduction), the firing frequency decreases [7,8,9]. This change in activity precedes the movement by approximately 20 ms. Once the new ocular position is reached, the firing frequency stabilizes, being higher the more the eye is displaced toward the activation direction. A similar dynamic is recorded in motoneurons of the oculomotor nucleus and the troclear nucleus for movements in the inward/outward and vertical (upper and lower) direction.

The pons reticular formation plays a crucial role in the control of eye movements. Lesions in this region have been observed to cause gaze paralysis [10]. Within this area, several neuronal populations which are related both to the generation of saccadic movements and to the maintenance of the eyes in a fixed position in space have been identified. Excitatory and inhibitory burst neurons have fast and phasic firing. These neurons are organized ipsilaterally and contralaterally with respect to the oculomotor nuclei to which they project [7,11,12]. These neurons are active during eye movements in all directions in space. Excitation and inhibition vary according to the direction of movement. Their main function is to generate the pulse necessary to activate the motoneurons and interneurons of the oculomotor nuclei, thus allowing the execution of eye movements.

A theoretical model for the generation of saccadic movements has been proposed in which excitatory burst neurons are activated when a saccade is to be made [12,13,14]. These neurons receive information from higher centers about the desired eye position, while they are inhibited by neural information of the eye position calculated by tonic neurons. It has been proposed based on neural network simulations that burst neurons would perform spatial–temporal transformations between the colliculus and the motoneurons [15].

The so-called omnipause neurons exhibit continuous firing in the resting state. However, during a saccade in either direction, they cease their activity until the eye movement concludes [16]. These neurons have inhibitory connections with both inhibitory and excitatory burst neurons [17,18]. In addition, their activity can be modulated by visual pathways from the superior colliculus (SC) and optic chiasm, which temporarily inhibits these cells [19]. Omnipause neurons have the function of inhibiting saccadic movements [16,18]. Its dramatic role can be observed by stimulation of omnipause neurons from the rostral tectum which induces its activation, inducing the maintenance of eye position by suppressing saccades, while the activation of the caudal tectum activates the inhibitory burst neurons which liberates the initiation of saccades by inhibiting omnipause neurons [20,21]. The saccade stop signal has been attributed not only to omnipause neurons but also to the fastigial nucleus [22].

The nucleus prepositus hypoglossi contains neurons with tonic, tonic-phasic and phasic discharge patterns [23]. It projects to the ocular motor nuclei and premotor structures such as the cerebellum and vestibular nuclei [24,25,26]. The tonic neurons maintain continuous and stable firing, the frequency of which is proportional to eye position. These cells play an essential role in maintaining the ocular position once the saccadic movement has been completed. It is proposed that collaterals of excitatory and inhibitory burst neurons connect to the neural integrator which would compute the eye-position signal to be maintained by tonic neurons [13,23,27]. Cerebellar lesion causes eye drift during fixations and is therefore considered to stabilize the “neural integrator” [28]. Not only prepositus hypoglossi presents tonic fixation neurons [29], but also frontal eye fields (FEFs), which would contribute to maintaining eye fixations by preventing reflexive saccades to other potential targets in the visual field.

The SC stimulation produces rapid eye movements [30,31,32,33,34], and excitation of oculomotor premotor areas [35,36,37]. The SC is considered an important hub for programming saccadic eye movements in which saccade direction is programmed vectorially in a retinotopic map. The strength of functional connectivity from SC to burst neurons is organized in a retinotopic manner [38,39]. This pattern of connection permits the transformation from a spatial retinotopic code in the SC into a brainstem temporal code (firing rate and duration) in burst neurons in order to drive extraocular muscles through the motoneurons projection. Saccadic eye movement can be generated without SC by means of connections between the FEF and brainstem. However, the SC integrates information from bottom-up saliency maps in the striate and extrastriate cortex, task requirements from FEF, reward previous history from the basal ganglia and substantia nigra, and homeostatic states from the zona incerta [reviewed in [40,41]].

### 1.2. Visual Position Selection

The top-down and bottom-up influences interact to generate eye saccadic movements [42]. Bottom-up factors influencing saccadic decisions include abruptly occurring stimuli [43], unique features that pop-out in visual search [44], high spatial frequency content and edge density [45], higher local spatial contrast [46], luminance contrast and edges [47], high spatial frequency edge information [48], color and others [49]. These results have allowed the proposal of the presence of a saliency map, a product of the integration of feature maps, which would drive the selection of certain fixation points of the visual scene [50]. Such saliency maps would interact with top-down processes for the target selection, with factors such as task demand [51] and semantic information [52].

As outlined above, the saccadic system is a very complex system, which on the contrary, when naturally inspecting in a free manner, the visual field responds to a relatively simple pattern. Here, we will focus on the temporal aspects, specifically the duration of fixations.

### 1.3. Eye Fixation Durations (EFDs)

During free viewing, subjects make around three to four saccades per second, no matter which kind of static image they are looking at [53], or when looking at a video [54]. The means of EFDs are modulated by experimental conditions such as luminance or type of image [55,56], but also by the characteristics of the next saccade and the experimental task [57,58,59]. However, not only first-order statistics such as mean and standard deviation are important, but also the probability density function (PDF) that fits the frequency histograms, because the parameters of the distribution would allow the analysis of changes in the parameter values related to the experimental conditions, such as type of images or repetition of images, but also the PDF would suggest some aspects of the internal dynamics that underlie the EFD, interrupted by saccades to new directions. One particular PDF which has been successfully applied to the duration times of fixations is the exGaussian distribution [60]. As indicated before, to have a PDF with certain parameters, such as those of the exGaussian (exponential function convolved with a Gaussian distribution to obtain the exGaussian; with parameters µ: mean; σ: standard deviation of the Gaussian; and τ: exponent of the exponential function) would allow the identification the possible relationship of these parameters with a defined cognitive processes, or on the contrary its common participation in a given cognitive process [60]. For instance, for the exGaussian model, more predictable words related to smaller and less lexical ambiguity are related to smaller µ values [61,62]. In the inspection of visual scenes, the exGaussian showed a better fit of EFD histograms than the Gaussian distribution [60], and the exGaussian parameters were related to specific characteristics of the EFD, such as the type of image changes in the µ Gaussian component, while familiarity was related to the exponential component τ. Although entropy metrics have been obtained for the eye fixation position distributions [56], in order to assess distribution spread, entropy has not been computed for EFD, and it would be one of the objectives of present report.

### 1.4. Competition Model (C Model)

The present report tries to frame the EFD in a C model between the saccadic and the fixation system, based on a mutual inhibition and a continuous competition for remaining in the current fixation or producing a saccade to a new location. The model would consider that after a saccade, there is a certain refractory period for producing a new saccade. The possibility and generality of such a model have been proven in different settings: eye rivalry, the perception of ambiguous figures [61] and the lever-press response in variable interval schedule of reinforcement [62]. The competition model assumes that the expression of a given percept (or response) occurs when the underlying neural network obtains an activity value higher than that of any other alternative network [61,62,63], as in the winner-take-all algorithm [64]. Eye rivalry and ambiguous percepts would correspond to the winning image representation, and the lever press would correspond to the competition between the lever-press response and any other possible alternative motor response. The model presents the particularity that once a perceptual representation, or just after a response has been made, there is a refractory period for a new perception to be installed or a new response to be produced. This refractory period being modeled by a sigmoid function. Once the asymptotic value of the sigmoid function is reached, a non-biased competition is established (see a more detailed description of the model in the method section). The C model also allows the modulation of the probability of a particular percept or response by top-down processes, such as for instance attention [61].

Therefore, the successful modeling of the EFD by a competition rule plus a saccadic refractory period would give some hints about the underlying dynamical processes related to maintaining a fixation or the induction of a saccade. It is important to consider here that the exGaussian model has been proven to be a good approximation for fitting the EFD histograms [60]. And for this reason, it would also be fitted to the EFD histograms in the present report. However, it is not the objective of the present report to compare different models, given that different methods for fitting exGaussian and the competition model would be used, but also because other alternative models such as the gamma function, the Poisson process and many others would potentially fit the EFD histograms.

A series of complex models which take into account characteristics of the saliency maps [50], the modeling by random walks for deciding the timing of the next saccade [65], the neural control of saccadic and eye fixation networks [66], or the information gained by stay or go to a new visual scene location [67], have been proposed recently to explain the EFD distributions. The detail of description of these models would be more related to the so-called algorithmic and implementation levels upon Marr’s levels of analysis [68]. The simple and parsimonious descriptive C model expressed above, and in the methods section does not try to be compared with such complex models, but shows that from a descriptive manner, the EFD distributions can be explained by a PDF with a fixed parameter (A) which defines the saccadic refractory period, and a non-stationary time-dependent ps parameter which would be related to the free competition between different positions of the image, once the refractory period for a new saccade has been overcome. In this sense, the model would be more related to the more abstract level of computation in Marr’s proposal [68]. The C model is able to dissociate the often-analyzed mean of eye fixation durations in these two parameters (A and ps), defining saccadic refractoriness and steady probability for a new saccade, but also taking into account that the competition between neural networks is one of the basic process driving EFD and would be the main asset of the present model.

The present report tries to show the compatibility of the C model to adjust the EFD histograms in a situation of free viewing of four different type of images presented in five blocks (see methods). These data would be obtained from an open database [69], which contains the EFD during free viewing of four type of images (nature, urban, fractals, and pink noise) during five successive blocks. The proposed hypotheses are that the frequency histograms of EFD are fitted by the C model composed of a (i) geometric PDF modeling the competition between the saccadic and the eye fixation systems, (ii) with the probability of making a saccade in a given time bin (ps) being modulated by a sigmoid to assure the stability of the image for a certain period of time after a saccade. This approach would show the dynamics of the very complex biological network of the eye fixation and saccadic-saliency network at a conceptual quantitative approach. The conceptual quantitative approach is granted by the very complex anatomic–physiological structure and dynamics of these networks, which makes a detailed computational model at the neuronal level highly difficult.

The model explicitly tests the hypothesis that after a saccade, a refractory period for new saccades occurs. Our working hypothesis is that if the sigmoid modulated by the A parameter across very different visual scenes is relatively similar across conditions, it would imply the presence of a post-saccadic refractory motor period. This motor post-saccadic refractory period would permit a deep visual analysis of the foveated region. On the other hand, a variable ps parameter across the different type of presented images would suggest that the probability of making a saccade would depend on the saliency across the scanned image.

## 2. Methods

Please note that in the Appendix A are the most important scripts (and their functions) used in present report (model simulation, testing the model, creating histograms) and a clear description of how to access the data, and organize them in the MDTM data matrix which is used for analysis. The script order in the Appendix A is suggested to be followed.

### 2.1. Database

The data analyzed in this study were obtained from an eye movement and eye fixation data set. This data set stores eye movement recordings and eye fixations from 23 published studies conducted at the Institute of Cognitive Sciences of the University of Osnabruck and the University Medical Center Hamburg-Eppendorf. The data can be obtained from https://datadryad.org/stash/dataset/doi:10.5061/dryad.9pf75 (accessed on 9 October 2024), and the general organization of the data are described in Wilmig et al., 2017 [69].

The data set is stored in an HDF5 file titled “*etdb_1.0.hdf5*”. This format is extensively used to store large volumes of information in a structured and efficient manner. Each file in the data set contains records organized into vectors that encode information about the fixations. For this analysis, the study entitled “Memory I” was selected. The data and results of this study correspond to the experiment 1 of Kaspar and König [56]. It contains the following specific information: subject, fixation coordinates on the x and y, fixation start and end expressed in ms, type of figure (nature, urban, fractals, pink noise), trial number, and block number. The analysis included data from 45 subjects aged between 18 and 48 years, all with normal or corrected-to-normal visual acuity. Before participating, all subjects signed a written consent form to participate in the experiment. During the experiment, participants viewed five blocks of 48 images each with four different categories: nature (12), urban (12), fractals (12) and pink noise (12). Each image was presented for 6 s in a random order. The total number of reported fixations was 179,473.

As indicated in Kaspar and König [56], eye tracking was recorded with the Eye Link II system located on a 21” Samsung SyncMaster 1100 CRT monitor (for Eye Link II, the adress is Ottawa, ON, Canada; for Samsung SyncMaster 1100 CRT monitor, the adress is Suwon, South Korea). The screen distance was 80 cm and the screen resolution was 1280 × 960 pixels. To facilitate free visual behavior, no headrest was used. Eye movements were recorded at a sampling frequency of 500 Hz. Before experimental data recording, subjects were required to perform saccadic movements at different fixation points that appeared on the screen in a randomized order, for eye-position calibration purposes.

The detection and characterization of saccadic movements was performed automatically by the eye-tracker, based on three measurements: eye movement of at least 0.1° with a velocity of 30°/s and an acceleration of at least 8000°/s. After the initiation of the saccadic movement, the minimum velocity of saccades was 25°/s and had to be maintained for at least 4 ms.

From the data extracted of the database of Wilmig et al., 2017 [69] we organized the fixation duration as a matrix (matrix name: MDTM) with five columns. Column 1: fixation duration. Column 2: subjects (1–45). Column 3: order of image in each subject (1–240). Column 4: block number (1–5). Column 5: figure type (nature, labeled as 7; urban, 8; fractals, 10; pink noise, 11). A total number of 179,473 fixations were analyzed. Each EFD group of fixation inside a block for the same type of images was collapsed and analyzed independently. Therefore, for each subject, 20 numerical series of EFD were analyzed in each of the 45 subjects (5 blocks × 4 types of figures). EFDs higher than 1.5 s were considered outliers and eliminated (total number of outliers = 783, percentage of the total = 0.43% of the total number of reported fixations).

### 2.2. Models

The 20 series of EFD (5 blocks × 4 types of images) per subject, were organized in frequency histograms with bins of 50 ms. Then, the C model was applied to the EFD histograms. The C model parameters were computed from a homemade function in matlab (MatlabR2024b) called from a script.

The C model for the saccadic-eye fixation systems assumes that some sort of competition exists between these two behaviors, without making at this point any hypothesis about the underlying neural mechanism (see the discussion section for that). For the C model, the expression of a specific behavior depends on taking a higher neural activity than the alternative behavior (Figure 1A), following a dynamic similar to the winner-take-all algorithm [64] and the geometric distribution (see Figure 1B,C, and Equation (1)). Then, in Equation (1) the probability (ps) would be the probability that the saccade-related network obtains a value greater than the eye fixation-controlling network in a time period (bins), and (1 ps) would be the probability that the eye fixation-related network would have an activity value greater than the network controlling saccadic movements in a time bin. For the sake of computation, these probabilities are computed in bins of 50 ms, the same duration as the time bins for the built-up EFD histograms. Then, the probability (ps) that a particular ocular fixation between two saccades (EFD) obtain a particular value between time t − 1 and time t follows the geometric distribution. The geometric distribution here is applied with the number of time bins (of 50 ms) needed for the saccade-related network to win the competition with the eye fixation-related network.f (t − 1 < EFD < t) = (1 − ps) ^(t − 1)^ × ps(1)

This equation implies that the probability of a fixation time obtaining a certain value (between t − 1 and t) depends on the occurrence of the saccade at t = 0 and at t, in this sense: f = probability of an inter-saccadic value to be between t − 1 and t (it can also be expressed as number of fixations by multiplying the PDF by the number of fixations, also in Equation (3)); ps = probability that the saccadic network wins; t = time base (number of time bins from previous saccade needed for the saccadic network to have a higher value than the eye fixation network); t − 1 = times the fixation network wins in a row and fixation is maintained (ps)^(t−1)^ = probability that the fixation network wins the competition with the saccadic network in t − 1 bins.

Equation (1) implies that the probability of the saccadic network winning the competition (ps) is constant during an eye fixation (or inter-saccadic time). The model can be modified to allow ps, initially set at ps = 0 for post-saccadic t = 0, to increase with time from the last saccade to an asymptotic value of ps. This possibility has been previously successfully assessed in other processes which are also based on a competition rule: The perception duration in eye rivalry and ambiguous figures [61], and the inter-response time of press-lever responses in variable interval schedule of reinforcement [62]. Then, ps would have a relative refractory period in which the probability that a new saccade occurs at a certain time (t) will be a function of time since the last saccade, taking into account that now the probability of the saccadic network is a function of time since the previous saccade (ps′(t)) (Figure 1D and Equation (2)). For the present C model, it is hypothesized that after a given saccade, there is an inhibition to a new move after a saccade has been made. For this reason, the probability that the saccadic saliency network wins the competition would be a function of the time elapsed from the previous saccade (ps′(t)), and would increase as time passes to its asymptotic value (ps):ps′(t) = (1/(1 + (e^(−t*A)+e2^)) × ps(2)

A = parameter to modulate the curvature of the sigmoid.

e^2^ is introduced into the sigmoid equation to have the origin at time zero (ending time of previous saccade).

It must be taken in account that the ps′ parameter dynamics as depicted in Figure 1D is just the mean value across time points, and it should be considered the mean of a stochastic process [63], which in some time points would permit that ps > (1 ps), and then a saccade would be triggered (Figure 1A).

Finally, the probability than an EFD obtains a given value between t − 1 and t under the C model would be (Figure 1E,F):f (t − 1 < EFD < t) = [(1 − ps′(t))^(t−1)^ × ps′(t)]/T(3)

The term T is introduced to normalize Equation (3), so that the area under the curve would approximate to 1. T is not computed analytically but numerically, by summing the area under the curve created by the numerator of Equation (3).

By changing the curvature parameters of the sigmoid (A), which expresses the dependence of the time elapsed from the previous saccade to the current one, and of the probability that the saccade-related network wins the competition at ps asymptotic levels, different curves can be obtained (See Figure 1E and Figure 1F, respectively). To estimate A and ps, the values of A and ps are changed systematically, and by correlation between the empirical histogram and Equation (3), the optimal A and p parameters are estimated. Once the parameters A and ps have been estimated, the level of significance of the fit between the empirical distribution of EFD and the C model are estimated by means of the Kolmogorov–Smirnoff goodness-of-fit test [70]. The values higher than 700 ms were collapsed in EFD histograms and in the model.

As the exGaussian model has proven to be a good approximation for fitting the EFD histograms [60], we have reproduced this model and checked its ability to fit the EFD histograms. For that, we used the *exgfit* matlab function [71] to compute the optimal parameters (µ, σ, and τ) of the exGaussian distribution, and then we followed a procedure of optimization of these parameters by changing iteratively the values of these parameters, in a range of +100 to −100 in 1 unit step from the mean values computed with the *exgfit* function [71]. The Kolmogorov–Smirnoff goodness-of-fit test [70] was applied to test the goodness of fit between the exGaussian model and the EFD frequency histograms. It is important to consider that as indicated in the introduction section, the different fitting procedures used for both distributions (C and exGaussian) do not permit do claims about superiority of fitting between different models. However, and as an approximation to this issue, the Akaike information criterion (AIC) [72] was applied to the 900 analyzed EFD histograms (45 subjects × 5 blocks × 4 type of images). The AIC takes into account the number of adjusted parameters to decide on the performance of different alternative models. For the application of AIC, the exGaussian distribution needed to estimate 3 parameters (mu, sigma, and tau), while the competition model needs to adjust the ps value (probability of making a saccade in a defined period of time), and the A parameter which model the refractory period after a saccade has been produced.

### 2.3. Statistical Analysis

To test the effects of the factors, block and type of image on the mean values of the EFD, the A and ps parameters of the C model, and ANOVA with these two factors was applied using the JASP 0.19.3.0 [73]. To test the possible linear relationships between the EFD, the peak time of the C model, the peak time of the EFD histogram, and the parameters A and ps of the C model, robust regressions were computed with the *fitlm* function of matlab.

### 2.4. Shannon Entropy

The Shannon Entropy (Kaspar and König, 2011 [56]) was computed from the 20 series of EFD (5 blocks × 4 types of images) per subject, organized in frequency histograms with bins of 50 ms. Equation (4) was used to compute the entropy.H = ∑ −p log2 p(4)
where p is computed on each histogram bin by the fraction (counts in bin i/sum of bin counts).

## 3. Results

Figure 2 shows the values of the EFD means for the five blocks and four type of images. The ANOVA showed that the effect of the block factor (F(2.282, 104.42) = 4.029; *p* = 0.017; Eta squared = 0.014), the type of image factor (F(1.309, 57.577) = 83.946; *p* < 0.001; Eta squared = 0.458), and the interaction type of image × block were significant (F(6.855, 301.606) = 2.243; *p* = 0.032; Eta squared = 0.007). The Bonferroni post hoc for the block factor showed that only the comparison of block 1 with block 4 presented a trend for significance (*p* = 0.061; block1 < block4). The post hoc analysis of the type of image factor showed that the pink-noise image presented a higher EFD than all the other type of images (*p* < 0.001, with higher EFD for the pink-noise images for the three comparisons). The urban images presented a shorter EFD than the nature and fractal images (*p* < 0.001) (EFD general pattern: urban < nature = fractals < pink noise). There was a complex pattern of interactions between the effects of the block and the images. Given that the purpose of present report is not related to the interpretation of the relationship between blocks and type of image, these interactions are included in Appendix A but they are not further explored.

Figure 3 shows the overlapping of the C and the exGaussian models on the EFD frequency histograms. Two additional subjects are displayed in the Appendix A. The results show that both models are compatible with explaining the EFD histograms. The good fitting of the two models for the EFD histograms can be observed in these figures. The Kolmogorov–Smirnov test of goodness of fit was applied to the 45 subjects, in the four type of images and five blocks (900 tests), applying 50 ms bins (collapsing data > 750 ms) in order to test the adjustment of the competition and exGaussian models to the EFD histograms. Only in 14 cases the exGaussian model did not significantly fit the EFD histograms. The C model fitted the EFD in all cases. When bins width was of 40 ms (collapsing data > 600 ms), only in one case the exGaussian model did not fit the data; the C model fitted all cases. When bin width was 30 ms (collapsing data > 450 ms), only in one case the histograms were not fitted by the exGaussian and the C model. Appendix A shows the Akaike information criteria values for the competition and the exGaussian models from the 50 ms bins. The C model showed lower AIC values than the exGaussian model (791 cases over 900 comparisons: 87.9%). However, as indicated in the methods, section one of the objective of present report is demonstrating the compatibility of both models for explaining the EFD, and given the difficulty of making appropriate comparisons between models due to the different methods used for fitting parameters in both models, no further explorations of this issue were made.

Figure 4A shows the mean values for the A parameter of the competition model. The ANOVA showed that only the effect of the block factor was significant (F[3.239, 142.505] = 5.30; *p* = 0.001; Eta squared = 0.031). The Bonferroni post hoc showed differences between block 1 and block 2 (*p* = 0.004), block 1 and block 3 (*p* = 0.021), and block1 and block5 (*p* = 0.034). The results suggest that lower values of A imply a higher refractory period to reach the steady value of the ps parameter; block 1 presented a longer time to reach the saturation of the C model ps parameter. The absence of an effect of the type of image indicates that the A parameter value was steady across the different images.

Figure 4B displays the mean values of the ps parameter for the C model. The block factor presented a trend for significance (F[3.019, 132.82] = 14.128; *p* = 0.073; Eta squared = 0.011), due to trend for significance of block 1 > block 2 (*p* = 0.067). The ANOVA showed that only the effect of the type of image was significant (F[1.879, 82.667] = 14.128; *p* < 0.001; Eta squared = 0.065). The Bonferroni post hoc showed differences between the ps parameter for the images: nature < urban (*p* < 0.001); nature < fractals (*p* = 0.047); nature > pink noise (*p* = 0.05); urban > pink noise (*p* < 0.001), fractals >pink noise (*p* = 0.002). The ps parameter results indicate that the probability of performing a saccade in a time period of 50 ms (the period used as bins in the frequency histograms of Figure 3) is lower for the pink-noise image (ps general pattern: pink noise < nature < urban = fractals).

Figure 5A shows the robust regression between the time to reach the peak time of the C model and the peak of the EFD histograms. The high R^2^ indicates good adjustment between the model and the empirical data for peak times. Figure 5B,C show the linear robust regression of the A and ps parameter with the peak time of the EFD histograms, respectively. The much higher R^2^ for the regression of the A parameter suggests that it is the A parameter of the C model which defines the refractory period of the EFD histograms. However, the role of the ps parameter in defining the peak time of the EFD histogram, in a more limited manner, is observed by the statistical significance of the A parameter vs. peak time of the EFD histogram residuals (obtained from Figure 5B), when regressed with the ps parameter (Figure 5D). The latter results suggest that the empirical peak time of the EFD histogram depends also on the parameter ps, advancing the peak time when ps is high, and inducing a time delay when the ps parameter is low.

Figure 6 shows the robust regression of the EFD mean with the A (6A) and ps (6B) of the C model. The higher R^2^ in the ps regression suggests that the ps parameter, indexing the probability of making a saccade in a given interval time, is more related than parameter A for defining the EFD. However, the inverse relationship between the A and ps parameter suggests that both parameters present a certain level of common variance, and therefore for defining the EFD (Figure 6C).

Figure 7 shows the values of the EFD Entropy for the five blocks and four types of images. The ANOVA showed significant effects of the block factor (F(2.174, 95.663) = 2.707; *p* < 0.001; Eta squared = 0.06), and the type of image factor (F(1.668, 73.378) = 11.539; *p* < 0.001; Eta squared = 0.427). The Bonferroni post hoc for the block factor showed that block 1 presented a lower entropy than all the other blocks (*p* < 0.003 for all comparisons), and block 2 presented lower entropy than block 5 (*p* < 0.002). The post hoc analysis of the type of image factor showed that the pink-noise image presented a higher entropy than all the other type of images (*p* < 0.001); the nature image presented higher entropy than urban (*p* < 0.001), and urban presented lower entropy than fractals (*p* < 0.001). The type of image entropy general pattern was then urban < nature = fractals < pink noise. This pattern was consistent when the data were organized in bins of 30, 40, and 50 ms, suggesting robustness in the entropy pattern when bin widths were changed (Appendix A).

## 4. Discussion

The present report is based on the same data presented in [56], which analyzed the mean of EFDs across blocks and type of images. The results obtained in the present report broadly replicated the original results, as expected. Kaspar and König (2011) [56] showed an increase in the EFD with the block, indicating an increase in EFD for successive presentations. In the present report, the block factor effect was significant for the EFD (block 1 presenting the shorter EFD across blocks), but Bonferroni post hoc only presented a significance trend in block 1 with respect to block 4. For the type of image, the same pattern of urban < nature = fractals < pink noise for the mean of EFDs was obtained, same as in [56]. The authors interpreted these results of increased duration in late presentation blocks as a reflection of a deeper scrutiny of the image in late blocks, while in early blocks the attentional scanning of the images was more related to a more superficial scanning of the image. With respect to the type of image, the obtained different durations should be related to the basic characteristics of the image such as color or complexity, although possible effects of top-down processes such as motivation, esthetics or others cannot be discarded.

The important point for the present report with respect to the EFD mean analysis, given that we did not try to deepen in the influence of the type of image and block on the EFD, but to test the compatibility of the C model for EFD, is that the general trend obtained for increases in EFD with block and the EFD pattern for the different type of images, is preserved in the present analysis when compared with the original analysis [56]. The small differences in the analysis are possibly due to the different methods used to discard outliers. Here, we established a very conservative threshold of 1500 ms in order to keep a high percentage of eye fixations to give robustness to the model fitting. The later procedure produced a higher number of accepted fixations than in Kaspar and König [56], which used two standard deviations as a limit. The threshold limit in the present report also produces higher EFD means than in the original work.

The importance of defining not only first-order statistics of the EFD data has been highlighted by Guy et al. [60]. They have indicated that the use of central or dispersion measures can only account for a general picture of the data distribution, given that similar values can be obtained with a very different structure of the data. The exGaussian model, in which a Gaussian distribution is convolved with an exponential distribution has been broadly applied to explain the EFD histograms. This model has been proven to be a good approximation to explain the EFD histograms [74,75,76]. The results of present report also showed that the exGaussian model is compatible with the empirical EFD data from the Kaspar and König report [56]. Furthermore, the exGaussian model has been successfully applied to explain EFD histograms in reading tasks [61,62]; in the inspection of visual scenes [60], permitting to characterize the exGaussian parameters to the type of image and familiarity. Kieffaber et al. [77] proposed the parameters defining the exGaussian would be related to different psychological processes, as µ to perceptual processes and τ to decision processes. The importance of the exGaussian parameters as metrics to be correlated with cognitive processes is with no doubt an interesting approach which would allow the association of the parameters with specific computational brain processes.

On the other hand, the structure of the C model [61,62,63] makes two very specific proposals with respect to the parameters characterizing it: (i) there is an asymptotic probability to the value of making a saccade in a given time bin (ps), but (ii) after a saccade there is a reduced probability of making a saccade, which is modeled by a sigmoid dependent of the A parameter. The C model was validated by the goodness of fit with the EFD histograms, but also by the very high correlation between the peak times of the model and the peak time of the EFD histograms. In the present report, the Akaike information criteria [72] assigns a better performance to explain the EFD to the C with respect to the exGaussian model. However, the different methods used to adjust the parameters of both models make it difficult to make an appropriate comparison between the two models. The main point for these comparisons in the present report is that given the very high frequency of cases in which both models fitted the EFD histograms, both are numerically compatible with explaining the EFD. However, we would discuss only the possible meaning of the C model, which is the main objective of the present report.

The A parameter of the C model showed a high correlation with the peak time of the EFD histogram and with the peak time of the histograms of the C model, indicating that A is responsible for modeling the refractory time period after a saccade. The A parameter showed a lower value in the block 1 which implies a longer time to reach the peak time in block 1. The main role of the time to reach the histogram peak rather than to the total duration of the EFD, is supported by the modest relationship between the A parameter and EFD. The ANOVA showed that the A parameter is relatively steady across blocks and type of images (see below for A as a possible index of a fixed motor refractory period). On the other hand, the ps parameter in the C model defines the probability of making a saccade in a time bin (50 ms in present report) once the transient refractory period for making a saccade is over. Its close relationship with EFD appears in a similar pattern (although inverted) of post hoc comparisons between EFD means and ps means across image types: the EFD general pattern is urban < nature = fractals < pink noise and the ps general pattern is as follows: pink noise < nature < urban = fractals). Furthermore, the ps high relationship with EFD, much higher that the relationship between the A parameter and the EFD, suggests its critical role in defining the EFD. A closer look shows that the residuals of the relationship between histogram peak time vs. A are related to ps, suggesting that the peak time of the histogram depends on both: primarily on the A parameter defining the refractory period, but modulated by ps defining the probability of making a saccade, or (1 ps) staying in the current position (see Figure 1F). The latter is important, because we would discuss later that the preferential time for looking at the current visual scene position (time until the peak of the EFD histogram), is related primarily to a relatively fixed A parameter, but also to a more variable ps parameter which would modulate the time to peak of EFD histograms.

As any model in which external data are interpreted from an internal model, which is not empirically and simultaneously measured, it can be defined as an inverse modeling problem. The latter argument implies that the empirical EFD could be explained by different internal models. Although not explicit metrics for saliency images, as proposed in the Itti–Koch model [50], the validity of ps as an index of competition between the probabilities to stay or to move is implicit in the construction of the model itself. Specifically, ps is the parameter governing fixation duration: it represents the probability that a saccade will occur at a given moment, whereas (1 ps) represents the probability of maintaining the current fixation. Thus, a higher ps (corresponding to higher saliency of regions other than the current fixation, as illustrated by the green line in Figure 1C) leads to shorter fixation durations. Indeed, ps is inversely related to observed eye-fixation durations, suggesting that when saliency is higher at other image locations, gaze is more likely to shift toward them. Additional information about the internal model would help in increasing its validity. Therefore, some comments should be made about the possibility that the A and ps parameter would be related to some neuroanatomical structures and neural dynamics of the oculomotor and saliency networks, that, although not empirically recorded here, would suggest a neural compatibility with the C model for explaining EFD.

The SC plays an essential role in defining the amplitude and direction but also for deciding when to move the eyes, equivalent to ending eye fixation, although other areas such as the FEF can also initiate saccades [38,39,40,41]. The SC activates the brainstem oculomotor plant by increasing the activity of burst neurons [12], and inhibiting the omnipause neurons [16,21]. The role of the SC for selecting the direction of movements under competitive conditions has been demonstrated in a two-choice experimental situation in which excitatory bias from FEF would allow for more activity in the positively biased colliculus, which would be selected for action through a winner-take-all process, producing contraversive eye movements [78]. Such a model could be generalized to human visual scanning and EFD distributions such as those presented in this report. One critical aspect of the presented C model is the need for inhibition in the area generating the saccade, something that occurs internally in the SC, but also from the projection to colliculus from the substantia nigra pars reticulata [79]. SC should receive information from the saliency maps [50]. The saliency map also implies a sort of center-surround inhibition in the different representational maps created from bottom-up information and modulated by top-down inputs representing task, motivation, and many other possible aspects. The different saliency maps representing visual features, such as luminance contrast, color, opponency, oriented edges, flicker, and motion, should be finally integrated for generating a priority signal that would control orienting behavior, and optionally in overt attention a saccadic movement. Saliency maps and priority maps have been described in several brain areas as V1, V4; lateral intraparietal area, FEF, dorsolateral prefrontal cortex and SC [reviewed in [80]]. If neuroanatomy, including the presence of inhibitory synaptic connection, can be considered compatible, the dynamic aspects are much more difficult to interpret from the current scientific literature. It can be suggested that the refractory period for making a new saccade after completion of the previous saccade should be due to an inhibition of the motor programs to produce a saccade, and on the other hand, the possibility that inhibition of the current fixated position in saliency map of the visual scene occurs, which is being progressively adapted, permitting other areas of the saliency maps to win the competition for gaze orienting in a winner-take-all type competition [61,62,64].

From the modeling to neural implementation, two different subprocesses can be suggested: a motor refractory period indexed by the A parameter, which is relatively steady, and a free competition period to generate a saccade which would be indexed by a more variable ps parameter, which is very dependent on the image type. However, the peak time of EFD histograms would also be influenced by the ps parameter, with higher ps values shortening the peak time and low values increasing the peak time. The presence of a refractory period in the immediate post-saccadic period has been previously reported and interpreted as result of a random exponential model triggering saccades in the so-called α phase of the model [81]. The refractory period has also been modeled by a Poisson process constrained by a Gaussian inhibition period (refractory period), and a follow-up rebound of the Poisson λ parameter [82]. Both model share with the C model the proposal of a relatively fixed refractory period duration, that would influence the relatively steady rate of saccades, as has also proposed for other perceptual systems with fixed sensory inputs [61]. One possible source of this refractory period would be related to the so-called saccadic inhibition phenomenon, in which a saccade is delayed if a peripheral target is presented. Saccadic inhibition has been proposed to relay in the post-saccadic activity in visual cortex, then rooted to transiently inhibit the oculomotor system (Amit et al., 2017) [82]. The main difference from the C model is for the phase of free competition, given that the two indicated models rely on random exponential or Poisson processes [81,82], while the C model relies on the neurophysiological plausible concept of push–pull competition with a dynamics similar to a winner-take-all mechanism (levels 3 and 2 of Findlay and Walker) [66]. Although the present model is still stochastic, as the ps parameter is the mean probability of making a saccade in a given time, and (1 ps) would be the mean probability to maintain fixation in the same period, it can be easily rooted in the widespread activation–inhibition processes of the brain. If the decision to make a saccade is finally mainly taking place in the retinotopic map, the where and when to look would rely on competition, not only in the colliculus but in different cortical areas related to oculomotor and saliency maps [40,41]. This competition would depend on possible top-down influences of the familiarity acquired during the experiment and from the characteristics of the images [56], something which would be captured by the parameter ps, whose value would depend on the features described in the introduction section [43,44,45,46,47,48,49].

Another point to be discussed is if the empirical EFD transient refractory period could be considered a fixed period, possibly related to post-saccadic motor inhibition with a very rigid time dynamic, or on the contrary, would be a consequence to a progressive adaptation to the current scrutinized visual scene position, which would be more variable and dependent of image characteristics. The post-saccadic motor inhibition hypothesis is suggested by the relatively low differences in values of the A parameter across image types, although the much more variable ps parameter is also influencing the peak time of the EFD histograms obtained empirically, justifying the variability in the peak time of the empirical EFD histogram and in the peak time of the model fitting. Those results suggest that the refractory period of empirically recorded EFD would occur as a consequence of both: a relatively fixed post-saccadic motor inhibition similar to that described for the saccadic inhibition phenomenon [82], plus an influence of the image features, which would delay or advance the empirical refractory period. The significant differences in the ps parameter across image types would suggest that the saliency of the image would influence the probability of inducing a saccade. Once the sigmoid defining the ps parameter saturates, the free competition at the saliency map would be allowed and would be modeled by a geometric distribution. The competition side of the C model, given the stochastic character of ps′, would permit the exploration of broad content in the observed image, and would not become stuck in a given position of the image.

The pattern of the entropy metrics of the EFD was similar to that of the mean EFD, with increasing entropy with blocks and a pattern of urban < nature = fractals < pink noise. Higher entropy is interpreted as a more spread distribution of EFD. Interestingly, the pattern for EFD is opposite to that obtained by Kaspar and König (2011) [56] when computing entropy for eye fixation positions during image scanning. The increase in entropy with blocks is the opposite to that obtained by Kaspar and König (2011) [56] for the eye positions on the image. They interpreted that early exploration of the image was more exploratory; our results complement the spatial approach with the temporal approach and suggest that early exploration phases once an image position has been selected follow a more fixed pattern for eye fixation durations when the images have been previously visually analyzed. Also, the same pattern of EFD entropy opposite to eye-position entropy occurs for pink noise, being the highest in EFD and the lowest for eye-position fixations. However, the lowest EFD entropy was urban, while the highest eye-position fixation entropy was in nature images. Globally, some sort of trade-off in the strategy for image scanning can be suggested, in which if high entropy is allocated to eye position, low entropy is allocated to EFD, and vice versa. Furthermore, the entropy of eye movement consistency during facial exploration, as analyzed using the Eye Movement Hidden Markov Model (EMHMM) [83], shows that higher consistency (i.e., lower EMHMM entropy) is associated with greater efficiency and accuracy in neural representation, as well as with better facial recognition performance. Following this line of reasoning, the lower entropy of eye fixation durations (EFDs) observed in the first block may reflect more focused attention during the early presentations of the images, whereas in later presentations, reduced attentional engagement may weaken the neural representations and require longer processing time for adequate encoding. The higher entropy observed for pink-noise images is likely related to their inherently noisy structure, which hampers the formation of accurate and efficient neural representations. Whether complexity of the image is somehow related to EFD entropy remains question for future studies, which would open the possibility of a certain relationship between the image characteristics and the processing pattern in the brain.

A possible limitation of the present report is that the original database [69] did not separate regular saccades from intrusion saccades [84]. There is no consensus on whether intrusion saccades are visually guided, and their occurrence frequency is relatively low (about 18 per minute, compared with 240 regular saccades per minute), suggesting only a marginal role in the present analysis.

## Figures and Tables

**Figure 1 entropy-27-01079-f001:**
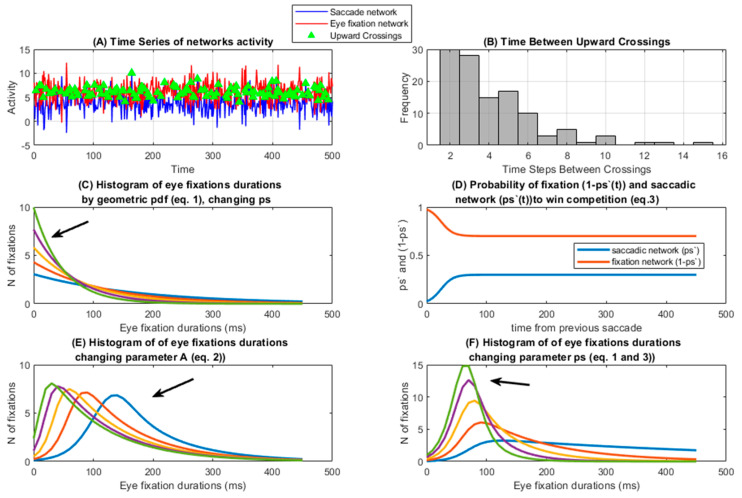
**Competition model.** (**A**) Simulation of the saccade and eye fixation network activity. The crossing points in which the saccade network presents more activity than the eye fixation network are labeled. (**B**) Frequency histogram of the time between two crossings of activity higher in the saccade than in the eye fixation network. (**C**) Probability density function (PDF) of a geometric distribution following Equation (1). The arrow indicates the PDF with the higher ps value. (**D**) Change in ps′(t) values: probability that the saccade network wins the competition in a given time bin, taking into account the time elapsed from the previous saccade (Equation (2)); (1 ps) represents the probability that the eye fixation network wins the competition. (**E**) Frequency histogram of fixation duration times as computed from Equation (3), obtained by changing the geometric distribution by modulating the A parameter, and consequently the ps′(t) (Equation (2)). The arrow indicates the PDF with the lower A value. (**F**) Same as (**E**) but changing asymptotic ps values and keeping fixed the A parameter. The arrow indicates the PDF with the higher ps value.

**Figure 2 entropy-27-01079-f002:**
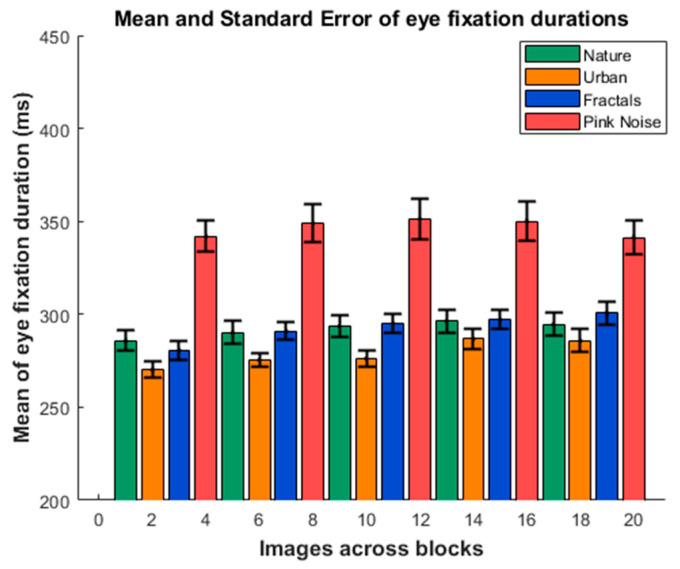
**Mean of eye fixation durations.** The image shows the mean and standard error of the eye fixation duration across the five consecutive blocks for the four presented type of images (nature, urban, fractal, and pink noise).

**Figure 3 entropy-27-01079-f003:**
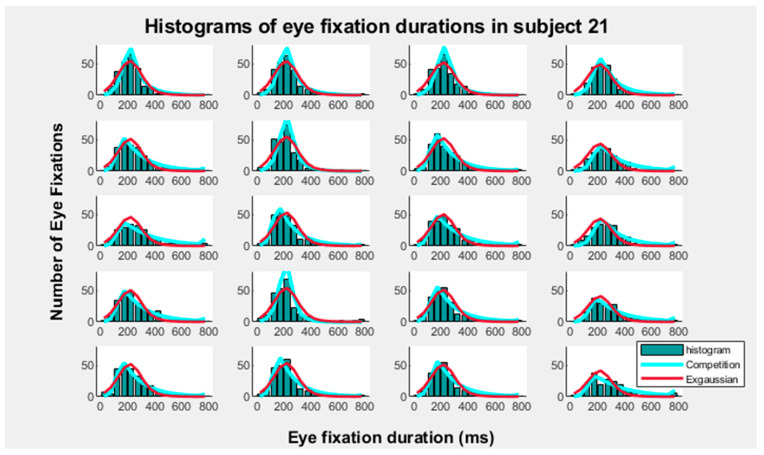
**Competition and exGaussian modeling of eye fixation duration histograms.** The image shows the fitting of the competition and the exGaussian models of the eye fixation duration frequency histograms for a single subject (subject 21). The fitting of the two distributions is displayed for the four type of images (columns) and for the five block of images presentations (rows).

**Figure 4 entropy-27-01079-f004:**
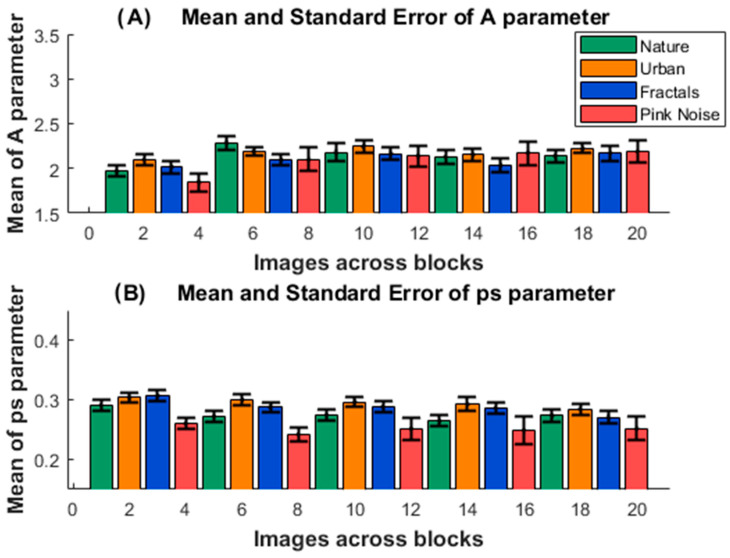
**A and ps values of the competition model.** (**A**). The image shows the mean and standard error of the A (**A**) and ps (**B**) parameters of the competition model, for the five consecutive blocks and for the four presented type of images (nature, urban, fractal and pink noise).

**Figure 5 entropy-27-01079-f005:**
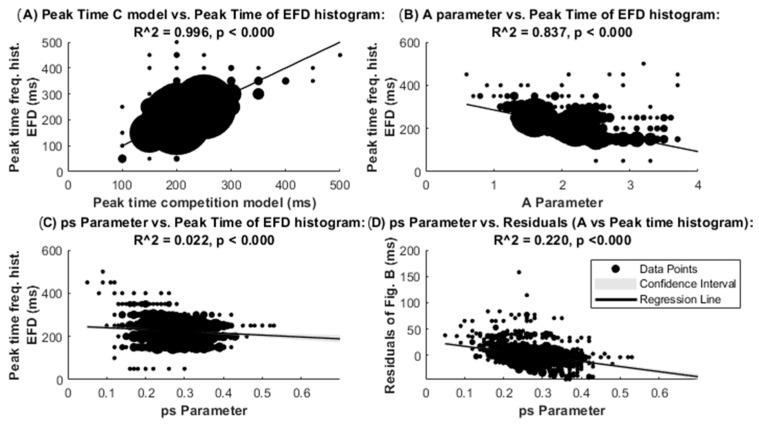
Robust regression between parameters and peak times of the competition model and peak times of the eye fixation duration histograms. Robust regression between the peak time of the empirical peak time of the histograms of the eye fixation durations and the peak times of the competition model (**A**), the A parameter (**B**), and the ps parameter (**C**). (**D**) shows the residuals of the regression in (**B**) vs. the ps parameter. The area of each point is proportional to the number of points with the same value. Confidence intervals of the regression are barely visible due to the high number of represented points (900). EFDs: eye fixation durations.

**Figure 6 entropy-27-01079-f006:**
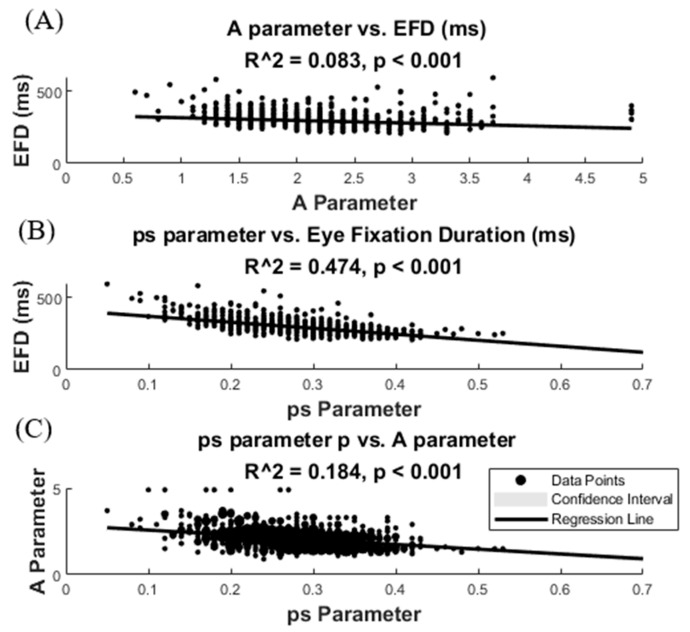
**Robust regression of the mean of eye fixation duration (EFD) with A and ps parameters of the competition model.** Eye fixation duration regression with the A (**A**) and ps (**B**) parameters. (**C**) displays the regression between parameters A and ps. Confidence intervals are barely visible due to the high number of represented points (900).

**Figure 7 entropy-27-01079-f007:**
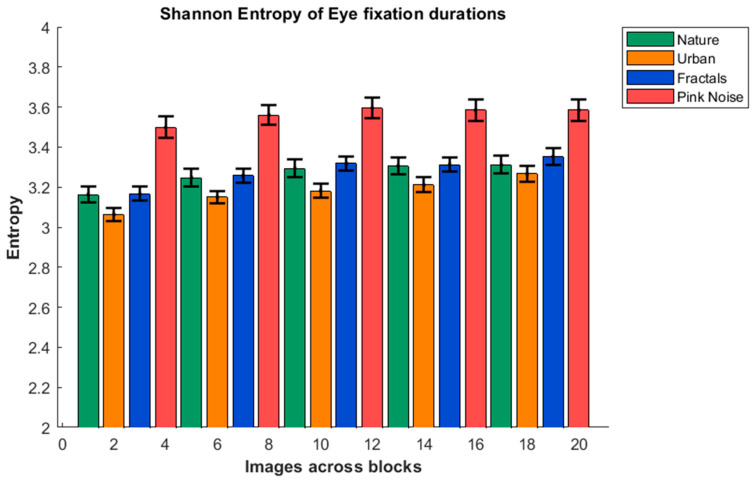
**Shannon Entropy of eye fixation durations.** The image shows the mean and standard error of the entropy of eye fixation duration across the five consecutive blocks for the four presented type of images (nature, urban, fractal, and pink noise).

## Data Availability

The data analyzed in this study were obtained from an eye movement data set. This data set stores eye movement recordings from 23 published studies conducted at the Institute of Cognitive Sciences of the University of Osnabruck and the University Medical Center Hamburg-Eppendorf. The data can be obtained from https://datadryad.org/stash/dataset/doi:10.5061/dryad.9pf75 (accessed on 9 October 2024), and the general organization of the data is described in Wilmig et al., 2017, [69]. The code can be found in the Appendix A.

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
