# Peer review of "Compatibility of a Competition Model for Explaining Eye Fixation Durations During Free Viewing"

_entropy, 2025, doi:10.3390/e27101079_

Round 1
Reviewer 1 Report
Comments and Suggestions for Authors
The authors present a novel approach to explaining eye fixation durations in healthy individuals. The article primarily focuses on fitting the equations of the C-model to an eye fixation database, while entropy itself is neither central to the study design nor to the discussion of the results. Therefore, it is questionable whether this interesting investigation is an ideal fit for the readership of Entropy.
In addition to this main limitation, I also note several minor issues:
Introduction
-
The relationship between saccade amplitude and peak velocity is linear only for small saccades. Please correct this.
-
In healthy individuals, not only microsaccades but also larger saccades known as square-wave jerks can occur (with up to 10 square-wave jerks per minute considered normal). This should also be taken into account and discussed in the Discussion section.
Methods
Since every fast movement greater than 0.1° was considered a saccade, all square-wave jerks are also included. This constitutes a limitation that should be discussed.
Results
In Figure 3, the histograms show some eye fixation durations around 100 ms or even shorter. How is this possible? Do the authors consider an intersaccadic interval shorter than 100 ms as normal, or is this an artifact of the detection algorithm?
Discussion
A paragraph addressing the study’s limitations is missing and should be added.
Author Response
The authors present a novel approach to explaining eye fixation durations in healthy individuals. The article primarily focuses on fitting the equations of the C-model to an eye fixation database, while entropy itself is neither central to the study design nor to the discussion of the results. Therefore, it is questionable whether this interesting investigation is an ideal fit for the readership of Entropy.
Thanks for your comment, with respect to the relationship of the Ms. with respect to the topics of Entropy, as a journal, I would like to comment that if the main goal of the report is about describing an abstract internal dynamics for the networks controlling saccadic eye movements, the computed entropy of eye eixation durations (EFD) give some insight, at least descriptive, of image scanning brain strategy. We have included some new comments in the discussion, including some suggestions from another reviewer, but we agreed that next studies on the topic should relate the image entropy with eye fixation patterns.:
The pattern of the entropy metrics of the EFD was similar to that of the mean EFD, with increasing Entropy with blocks and a pattern of urban<nature=fractals<pink noise. Higher Entropy is interpreted as a more spread distribution of EFD. Interestingly, the pattern for EFD is opposite to that obtained by Kaspar and König (2011) when computing Entropy for eye fixation positions during image scanning. The increase of entropy with blocks is the opposite to that obtained by Kaspar and König (2011), for the eye positions on the image. They interpreted that early exploration of the image was more exploratory, our results complement the spatial approach with the temporal approach and suggests that early exploration phases once an image position has been selected follows a more fixed pattern for eye fixation durations, that when the images has been previously visually analyzed. Also, the same opposite EFD entropy pattern to eye position entropy occurs for pink noise, being the highest in EFD and the lowest for eye position fixations However, the lowest EFD entropy was urban, while the highest eye position fixations entropy was in nature images. Globally, some sort of trade-off in the strategy for image scanning can be suggested, in which if high entropy is allocated to eye position, low entropy is allocated to EFD, and viceversa. Furthermore, the entropy of eye movement consistency during face exploration, as analyzed using Eye Movement Hidden Markov Model (EMHMM) [83], shows that higher consistency (i.e., lower EMHMM entropy) is associated with greater efficiency and accuracy in neural representation, as well as with better face recognition performance. Following this line of reasoning, the lower entropy of eye fixation durations (EFD) observed in the first block may reflect more focused attention during the early presentations of the images, whereas in later presentations, reduced attentional engagement may weaken the neural representations and require longer processing time for adequate encoding. The higher entropy observed for pink-noise images is likely related to their inherently noisy structure, which hampers the formation of accurate and efficient neural representations. If complexity of the image is somehow related to EFD entropy, remains for future studies, which would open the possibility of a certain relationship of the image characteristics with the processing pattern in the brain.
In addition to this main limitation, I also note several minor issues:
Introduction
-
The relationship between saccade amplitude and peak velocity is linear only for small saccades. Please correct this.
We have modified this sentence:
“They have linear relationship between the amplitude and duration of the saccades in the range between 1o to 50o [1,2,3], as well as between the maximum velocity and the amplitude of the movement, up to saccades with amplitudes of about 15°, at which eye velocity slowly saturates. [4].”
-
In healthy individuals, not only microsaccades but also larger saccades known as square-wave jerks can occur (with up to 10 square-wave jerks per minute considered normal). This should also be taken into account and discussed in the Discussion section.
We have included the following text as a limitation.
A possible limitation of the present report is that the original database [69] did not separate regular saccades from intrusion saccades [84]. There is no consensus on whether intrusion saccades are visually guided, and their occurrence frequency is relatively low (about 18 per minute, compared with 240 regular saccades per minute), suggesting only a marginal role in the present analysis.
Methods
Since every fast movement greater than 0.1° was considered a saccade, all square-wave jerks are also included. This constitutes a limitation that should be discussed.
As indicated before.
Results
In Figure 3, the histograms show some eye fixation durations around 100 ms or even shorter. How is this possible? Do the authors consider an intersaccadic interval shorter than 100 ms as normal, or is this an artifact of the detection algorithm?
We have followed completely the database of Wilmig et al, reference 69. Possibly its algorithm permitted the detection of very short fixation durations, as for instance those on double saccades. The Figure 1b of the reference 69 of Wilmig et al. (2011), in which not only the durations relative to our analysis are plotted, but also the other 22 studies described in this Wilmig et al. article, is very clarifying on this issue. Please notice that our distributions are pretty similar, including short eye fixation durations, to the 23 studies displayed by the authors which collected the data and appears in the figure 1b of the article describing the data base (Wilmig et al., 2011; Please recover the Fig 1b from : https://www.nature.com/articles/sdata2016126). Therefore, we have tried not to bias the results more than the authors of the eye movement recordings.
Discussion
A paragraph addressing the study’s limitations is missing and should be added.
The paragraph above as a limitation has been included.
“A possible limitation of the present report is that the original database [69] did not separate regular saccades from intrusion saccades [84]. There is no consensus on whether intrusion saccades are visually guided, and their occurrence frequency is relatively low (about 18 per minute, compared with 240 regular saccades per minute), suggesting only a marginal role in the present analysis.”
Reviewer 2 Report
Comments and Suggestions for Authors
The paper proposes a two-parameter competition model for free viewing, combining a post-saccadic refractory component governed by a sigmoid with curvature A, and a discrete-time competition component with probability pₛ that a saccade wins, yielding a geometric-like distribution for EFD. The model is fit to 179473 fixations from 45 participants viewing four image categories across five blocks; an ex-Gaussian is used as a comparator. The authors report that both models fit well by K–S tests, with the C-model showing lower AIC in 88% of histograms; A tracks the histogram peak timing while pₛ tracks mean EFD and image-dependent saliency.
Major concerns:
- The time-varying hazard is defined via a custom sigmoid with an added constant to set the origin and the pmf is then renormalized numerically by an empirical factor T. This lacks a principled probabilistic derivation and obscures parameter meaning; a standard discrete-time hazard/survival formulation is needed.
- All analyses use 50-ms bins and values >700 ms are collapsed; EFDs >1.5 s are discarded. Robustness to bin width and tail treatment should be reported, as these choices directly affect parameter estimates and GOF metrics.
- The reading of ps as a saliency/competition index is plausible but indirect. It should be corroborated by correlations with independent saliency metrics rather than ANOVA patterns alone.
- Entropy is computed on 50-ms histograms; its value is not scale-invariant.
- Discuss how this study could affect the eye-movement studies, such as "Understanding the role of eye movement pattern and consistency during face recognition through EEG decoding"?
Author Response
The paper proposes a two-parameter competition model for free viewing, combining a post-saccadic refractory component governed by a sigmoid with curvature A, and a discrete-time competition component with probability pₛ that a saccade wins, yielding a geometric-like distribution for EFD. The model is fit to 179473 fixations from 45 participants viewing four image categories across five blocks; an ex-Gaussian is used as a comparator. The authors report that both models fit well by K–S tests, with the C-model showing lower AIC in 88% of histograms; A tracks the histogram peak timing while pₛ tracks mean EFD and image-dependent saliency.
Thanks for the time dedicated to review our Ms,, and we hope to improve its quality following your observations.
Major concerns:
-
The time-varying hazard is defined via a custom sigmoid with an added constant to set the origin and the pmf is then renormalized numerically by an empirical factor T. This lacks a principled probabilistic derivation and obscures parameter meaning;a standard discrete-time hazard/survival formulation is needed.
We think to the reviewer to this very interesting suggestion. However, at this point we are not interested in developing a formal probabilistic account for the EFD histograms, but to test in a more functionalistic way if a model in which neural networks are competing for overt behavior (eye fixations or saccades), can be supported. The development of a formal theory would also imply quite a few requirements as each realization being a Bernouilli process or timing of the events to be continuous. Our present formulation is more biologically routed and not make any sort of a priori characteristics of the events, and follows the logic of statistics rather than formal probabilistic theory.
-
All analyses use 50-ms bins and values >700 ms are collapsed; EFDs >1.5 s are discarded. Robustness to bin width and tail treatment should be reported, as these choices directly affect parameter estimates and GOF metrics.
To test the possible dependency of the GOF metrics on bin width, we varied the bin size (30, 40, and 50 ms) and the upper tail collapse thresholds (450, 600, and 750 ms). We chose to reduce rather than increase the bin width in order to enhance temporal discriminability. The results showed very similar values for the number of histograms that could be successfully fitted, indicating a high robustness of the GOF metrics to variations in both bin width and tail-collapse parameters. This analysis is now included in the Results section.
“The good fitting of the two models for the EFD histograms can be observed in these fig-ures. The Kolmogorov-Smirnov test of goodness of fit was applied to the 45 subjects, in the 4 type of images and 5 blocks (900 tests) applying 50ms bins (collapsing data >750ms), in order to test the adjustment of the competition and exgaussian models to the EFD histograms. Only in 14 cases the exgaussian model did not significantly fitted the EFD histograms. The C model fitted the EFD in all cases. When bins width was of 40ms (collapsing data >600ms) only in one case the exgaussian model did not fit the data, The C-model fitted all cases. When bins width was 30ms(collapsing data >450ms) only in one case the histograms were not fitted by the exgaussian and the C-model.”
-
The reading of ps as a saliency/competition index is plausible but indirect. It should be corroborated by correlations with independent saliency metrics rather than ANOVA patterns alone.
We agree with the reviewer that interpreting ps as a measure of competition between networks representing different image positions (i.e., saliency in psychophysical terms) is somewhat speculative. We have not employed any formal computational saliency model, such as the Itti–Koch model, to quantify the saliency of our images. However, the validity of ps as an index of competition between the tendencies—or probabilities—to stay or to move is implicit in the construction of the model itself. Specifically, ps is the parameter governing fixation duration: it represents the probability that a saccade will occur at a given moment, whereas (1 – ps) represents the probability of maintaining the current fixation. Thus, a higher ps (corresponding to higher saliency of regions other than the current fixation, as illustrated by the green line in Figure 1C) leads to shorter fixation durations. Indeed, ps is inversely related to observed eye-fixation durations, suggesting that when saliency is higher at other image locations, gaze is more likely to shift toward them (middle part of figure 6) . We have now included this clarification in the revised manuscript.
“As any model in which external data are interpreted from an internal model, which is not empirically and simultaneously measured, it can be defined as a modeling inverse problem. The latter argument implies that the empirical EFD could be explained by different internal models. Although not explicit metrics for saliency images, as proposed in the Itti-Koch model [50], the validity of ps as an index of competition between the probabilities to stay or to move is implicit in the construction of the model itself. Specifically, ps is the parameter governing fixation duration: it represents the probability that a saccade will occur at a given moment, whereas (1 – ps) represents the probability of maintaining the current fixation. Thus, a higher ps (corresponding to higher saliency of regions other than the current fixation, as illustrated by the green line in Figure 1C) leads to shorter fixation durations. Indeed, ps is inversely related to observed eye-fixation durations, suggesting that when saliency is higher at other image locations, gaze is more likely to shift toward them. Additional information about the internal model would help in increasing its validity. Therefore, some comments should be done about the possibility that the A and ps parameter would be related to some neuroanatomical structures and neural dynamics of the oculomotor and saliency networks, that although not here empirically recorded would suggest a neural compatibility with the C model for explaining EFD.
The SC plays an essential role…”
-
Entropy is computed on 50-ms histograms; its value is not scale-invariant.
We have conducted a sensitivity analysis and showed the robustness of the entropy results, now in the main text and in the new supplementary figure 4. Bin width were reduce by the reason described in comment 2. Now in the text:
“The type of image entropy general pattern was then urban<nature=fractals<pink noise. This pattern was consistent when the data were organized in bins of 30, 40 and 50 ms, suggesting robustness in the entropy pattern when bins width were changed (Suppl. Fig. 4).”
-
Discuss how this study could affect the eye-movement studies, such as "Understanding the role of eye movement pattern and consistency during face recognition through EEG decoding"?
The very interesting report suggested by the reviewer has been included in the paragraph devoted to the entropy metrics in the Discussion section, together with some suggestion from the other reviewer:
“The pattern of the entropy metrics of the EFD was similar to that of the mean EFD, with increasing Entropy with blocks and a pattern of urban<nature=fractals<pink noise. Higher Entropy is interpreted as a more spread distribution of EFD. Interestingly, the pattern for EFD is opposite to that obtained by Kaspar and König (2011) when computing Entropy for eye fixation positions during image scanning. The increase of entropy with blocks is the opposite to that obtained by Kaspar and König (2011), for the eye positions on the image. They interpreted that early exploration of the image was more exploratory, our results complement the spatial approach with the temporal approach and suggests that early exploration phases once an image position has been selected follows a more fixed pattern for eye fixation durations, that when the images has been previously visually analyzed. Also, the same opposite EFD entropy pattern to eye position entropy occurs for pink noise, being the highest in EFD and the lowest for eye position fixations However, the lowest EFD entropy was urban, while the highest eye position fixations entropy was in nature images. Globally, some sort of trade-off in the strategy for image scanning can be suggested, in which if high entropy is allocated to eye position, low entropy is allocated to EFD, and viceversa. Furthermore, the entropy of eye movement consistency during face exploration, as analyzed using Eye Movement Hidden Markov Model (EMHMM) [83], shows that higher consistency (i.e., lower EMHMM entropy) is associated with greater efficiency and accuracy in neural representation, as well as with better face recognition performance. Following this line of reasoning, the lower entropy of eye fixation durations (EFD) observed in the first block may reflect more focused attention during the early presentations of the images, whereas in later presentations, reduced attentional engagement may weaken the neural representations and require longer processing time for adequate encoding. The higher entropy observed for pink-noise images is likely related to their inherently noisy structure, which hampers the formation of accurate and efficient neural representations. If complexity of the image is somehow related to EFD entropy, remains for future studies, which would open the possibility of a certain relationship of the image characteristics with the processing pattern in the brain. ”
Round 2
Reviewer 1 Report
Comments and Suggestions for Authors
no more comments
Reviewer 2 Report
Comments and Suggestions for Authors
The authors addressed all my concerns. Thank you.